# Not by the light of the moon: Investigating circadian rhythms and environmental predictors of calling in Bornean great argus

Dena J. Clink[1]*, Tom Groves[2], Abdul Hamid Ahmad[3], Holger Klinck[1]

**1** Center for Conservation Bioacoustics, Cornell Laboratory of Ornithology, Cornell University, Ithaca, NY, United States of America, **2** School of Physics and Astronomy, University of St. Andrews, Scotland, United Kingdom, **3** Faculty of Sustainable Agriculture, University Malaysia Sabah, Sabah, Malaysia

* dena.clink@cornell.edu

## Abstract

Great argus pheasants are known for their elaborate visual mating displays, but relatively little is known about their general ecology. The use of passive acoustic monitoring—which relies on long-term autonomous recorders—can provide insight into the behavior of visually cryptic, yet vocal species such as the great argus. Here we report the results of an analysis of vocal behavior of the Bornean great argus (*Argusianus argus grayi*) in Sabah, Malaysia, using data collected with 11 autonomous recording units. Great argus regularly emitted two call types, the long call and the short call, and we found that although both call types were emitted throughout the day, the short calls were more likely to occur during the morning hours (06:00–12:00LT). Great argus were less likely to call if there was rain, irrespective of the time of day. A substantial portion of calls at our site (~20%) were emitted between the hours of 18:00–06:00LT. We found that for nighttime calls, calling activity increased during new moon periods and decreased during periods of rain. We attribute the negative influence of rain on calling to increased energetic costs of thermoregulation during wet periods, and propose that the influence of the lunar cycle may be related to increased predation risk during periods with high levels of moonlight. Little is known about the behavioral ecology of great argus on Borneo, so it is difficult to know if the results we report are typical, or if we would see differences in calling activity patterns depending on breeding season or changes in food availability. We advocate for future studies of great argus pheasant populations using paired camera and acoustic recorders, which can provide further insight into the behavior of this cryptic species.

## Introduction

Acoustic communication is a fundamental component of social interaction across taxa, and serves a myriad of functions including mate attraction, resource acquisition and recognition of conspecifics [1]. Given the importance of acoustic communication to understanding social behavior, there has been a substantial amount of work done investigating the intrinsic (*e.g.*,

**Data Availability Statement:** R code and data needed to recreate all analyses are available at https://github.com/DenaJGibbon/Calling-in-Bornean-great-argus.

**Funding:** The Fulbright ASEAN Research Award for U.S. Scholars (no award number given) provided funding for this work.

**Competing interests:** The authors have declared that no competing interests exist.

reproductive status [2, 3]) and extrinsic factors (*e.g.*, environmental variables [4]) that lead to variation in vocal behavior, with a particular emphasis on the dawn chorus [5–8]. Investigating the social and environmental influences on calling behavior is crucial for understanding the function(s) of acoustic signals. In habitats where vision is limited, such as tropical forests, animals often rely on long range acoustic signals for communicating with conspecifics, and animals are predicted to call at times when the signal can effectively travel over long distances [9]. For example, birds and nonhuman primates call less when it rains, which has been attributed to a reduction in communication space [10, 11]. On a more basic level, investigating spatial and temporal variation in calling behavior can provide insights into basic ecology and activity patterns of understudied animals.

Nocturnal calling behavior has been documented in a variety of North American bird species (18 out of 22 orders examined), with over 70% of night vocalizing birds being classified as diurnal [12]. There is substantial variation across species, populations and individuals in nocturnal calling, with some species regularly calling at night and others calling only rarely; these differences are attributed to presumably different functions of night calling [13]. In some cases (*e.g.*, robins *Erithacus rubecula*), nocturnal and diurnal song may serve similar functions [14]. Other proposed functions of nocturnal song include attracting migrating females [15], limiting acoustic competition or masking [16], and reducing predation risk [17]. Night calling may also be a response to increase in natural [5] or artificial light [18], or anthropogenic noise [19]. Although the presence of nocturnal singing has been documented across diverse avian taxa, there is little information about the patterns and functions of nocturnal song, and it remains a relatively poorly understood phenomenon [17].

Passive acoustic monitoring (PAM) is an important tool that can provide information on cryptic, yet vocal species, and there has been a substantial increase in terrestrial PAM applications in recent years [20]. Importantly, PAM can be used to provide important insights into behavioral patterns of animals that are active at night [21]. In tropical forest environments where vision is obstructed by dense foliage, acoustic monitoring can be more effective than other methods which rely on human observers [22]. In addition, PAM allows for temporal and spatial coverage that is not generally not possible using traditional methods that rely on human observers [23]. To-date, the majority of terrestrial PAM studies have focused on bats (50%), with only 20% focusing on birds, and the majority of all terrestrial studies–irrespective of taxa–have been conducted in northern temperate regions (65% [20]).

PAM can be used to study birds with a variety of research goals, including searching for presumed extinct species (*e.g.*, the ivory-billed woodpecker [24]) occupancy modeling [25], investigating activity patterns [26] and timing of migrations [27], along with monitoring of territorial dynamics of individuals [28]. In addition, there has been increasing interest in the use of PAM as a tool to monitor avian diversity in a variety of habitats [29–32]. Ornithologists have relied on acoustic data for decades, but PAM offers a permanent archive of these sounds, and also allows researchers to collect data continuously over 24-hour periods for weeks or months at a time, which can provide important insights into diel calling behavior and general activity patterns of focal taxa. A recent study used PAM to model the occurrence of crested argus in Song Thanh Nature Reserve, Vietnam, indicating that there is increasing interest in the use of PAM to provide important information about threatened, vocal species in Southeast Asia [33].

Borneo is a hotspot of biodiversity, and is home to over 620 species of birds [34]. The majority of studies of birds on Borneo have relied on human observers [35–42], despite the potential for PAM to improve monitoring efforts of Bornean birds, and the fact that in some cases autonomous recorders outperform human observers [43]. An important caveat is that reliance on human observers in some cases was related to the fact that birds were more easily observed visually than aurally (*e.g.*, [38]), and for the earlier studies PAM methods were not yet widely available. Recent

studies have utilized PAM to study vocal animals and their habitats on Borneo. For example, PAM was used to study calling behavior of Bornean gibbons and found that gibbons are less likely to call when it rains, and that males start their morning solos later if there was rain the night before [11]. PAM has also been used to study Bornean orangutan behavior [44, 45], and to quantify differences across soundscapes in Indonesian [46] and Malaysian Borneo [47].

Great argus pheasants (*Argusianus argus;* hereafter great argus) are among the largest pheasants in the world [48], and males create 'dancing grounds' in which they clear the forest floor and perform 'nuptial dances' for females [49]. They are divided into two subspecies: the Bornean great argus (*A. argus grayi*), which primarily inhabits the understory of rainforests on the island of Borneo, and the Malay-Sumatra great argus (*A. a. argus*), which are found on the Malay peninsula and the island of Sumatra. The species has a polygynous mating system, with males advertising on their dancing sites in an exploded-lek system [50, 51]. The mating season for argus pheasants on Borneo is not known, although in captivity females have laid clutches of two eggs anywhere from 14–17 times per year and during every month, indicating that it is possible for females to lay multiple clutches per year in the wild [51, 52]. Argus pheasants appear to lack a distinct calling season, as they can be heard throughout the year, but there have been documented differences in vocal activity from year to year [53]. Great argus pheasants are classified as 'near threatened' by the International Union for the Conservation of Nature [54] and populations are decreasing across their range.

Although great argus are well known for their elaborate visual displays [50, 55], relatively little is known about their vocal behavior. The majority of calls are emitted by males, although females will emit calls on occasion [50]. Here, we report one of the first analyses of vocal behavior of the great argus pheasant (*A. a. grayi*) using acoustic data collected with 11 autonomous recording units in Danum Valley Conservation Area, Sabah, Malaysia. Despite their size and loud distinctive calls great argus are difficult to observe in the wild. There have only been a few systematic studies of their behavior (*e.g.* [48, 52]), but there have been studies which relied on human observers to detect calling argus and estimate population density (*e.g.* [53, 56, 57]). The goals of the present study were to: 1) investigate temporal and spatial variation in great argus calling at our site; 2) determine if calling behavior was influenced by environmental variables such as temperature or rainfall; 3) investigate the influence of lunar cycle for calls that occurred between 18:00–06:00LT; and 4) test whether there were differences in patterns of use of two distinct call types known to be emitted by great argus (the long call and short call). Great argus long calls have been proposed to serve a territorial function, whereas the short calls may be used for mate attraction [50].

## Materials and methods

### Study design

We deployed 11 Swift autonomous recording units [58] on a ~750 m grid from February to July 2018 in Danum Valley Conservation Area, Sabah, Malaysia (N4.964936˚, E117.805116˚). See Fig 1 for a map of the recorder locations. The climate of Danum Valley Conservation has been described as aseasonal [59] so there were no *a priori* reasons to believe that seasonal variation in climatic variables would have an influence on great argus vocal behavior. We attached the recorders to trees at a height of approximately 2 meters above ground and recorded continuously for the duration of the study, or until battery and/or unit malfunction. We recorded at a sampling rate of 16 kHz, 16-bit resolution and a gain setting of 40 dB (mono, WAV format). The detection distance of the calls using these settings is unknown as there are not any reported source levels for this species. Detection distance for gibbon female calls which are relatively stereotyped, and have a frequency range to similar to great argus (500–1800 Hz [60])

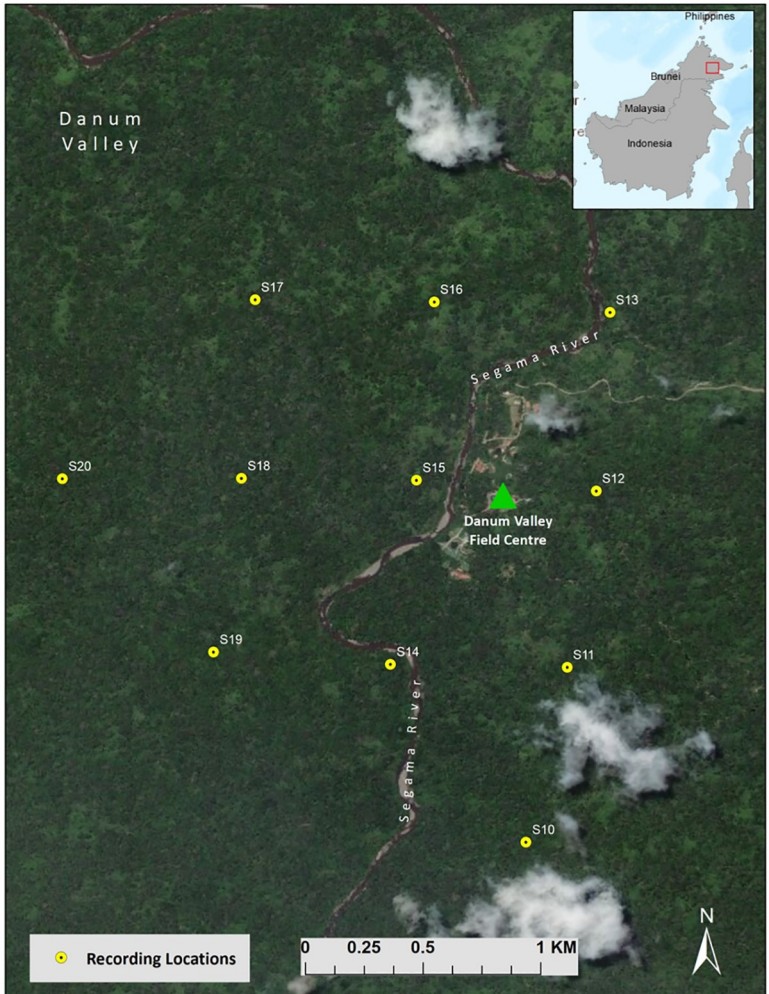

**Fig 1. Map of Swift autonomous recorder locations in Danum Valley Conservation Area, Sabah, Malaysia.** The map was made using ArcGIS (ESRI) v. 10.5.1 (www.esri.com).

was shown to be ~400 m at this site using the same recording settings [61]. Based on the subjective source levels of gibbons (which are one of the loudest acoustic signals in the environment) relative to great argus it seems unlikely that the detection range of great argus would be much larger than that of gibbons, which means that it was unlikely that we recorded the same great argus call at the same time on two different recorders.

Population density of great argus calling males in Sumatra was reported to be around 2.50 individuals per km$^2$ [52], whereas density of calling males in pristine forest in East Kalimantan, Indonesia was estimated to be 1.99 calling males per km$^2$ [56]. Our grid of autonomous recorders encompassed ~2.25 km$^2$, with the effective listening area being slightly larger. Therefore, based on the previously reported population density estimates from other sites, our study area probably encompassed the territories of anywhere from 4–10 calling great argus males.

## Acoustic analysis

Great argus males have been documented to have three distinct hooting calls: long calls, short calls, and irregular hoots. As described by Davison [50] in the Malay-Sumatra subspecies, the

long call is "a series of 15–72 hoots, beginning as monosyllables but the last half dozen progressively rising in pitch and becoming disyllabic." The short call is "a high-pitched hoot disyllable, kau-wow, the second syllable slightly the higher, lasting just under a second." The irregular hoot is "a series of two or three disyllables in which the first syllable is higher than the second, sounding like a series of inverted short calls . . . declining in pitch and speed." Calls may evoke a response of further calls from other males, and females occasionally give long calls [50].

To identify argus calling events in our long-term dataset, we used the Matlab-based acoustic analysis program Triton [62] to create long-term spectral average plots (LTSAs). We used temporal and spectral resolution settings of Δt = 300 s and Δf = 100 Hz. A single observer (DJC) identified all instances of argus calling in the LTSAs using a combination of aural and visual inspection of 24-hour periods for all hours of recordings in our dataset. We did our initial inspection of LTSAs over 24-hour periods as it allowed us to quickly scan through long time periods of acoustic data and identify calling bouts in the frequency range of interest (500–2000 Hz). Once we identified a calling bout in a 24-hour period, we then created short duration spectrograms to determine if the calling bout was great argus or a gibbon (as gibbon calling bouts appear similar on the 24-hour LTSA [11]); if it was a great argus call we then classified it as either a short call or a long call. Representative LTSAs of a 10-day period and a 24-hour period are shown in Fig 2A and 2B, and representative spectrograms of a short call and long call are shown in Fig 2C and 2D. We did not use the 10-day LTSA for detecting great argus calling events, but we include a 10-day LTSA in Fig 2 to illustrate how LTSAs can be used to investigate patterns of calling at various temporal levels. Although Davison [50] described three distinct call types, we found it difficult to distinguish between the short call and the irregular hoot in this population, so we included both short calls and irregular hoots in the short call category. Therefore, our analyses are based on two call types: the long call and the short call.

## Statistical analysis

**Environmental predictors of great argus calling events.** We investigated differences in argus calling events across 24-hour periods, and tested whether argus calling could be predicted by broad-scale environmental variables including temperature (degree Celsius; minimum and maximum in a 24-hour period), amount of rainfall (mm) in a 24-hour period, presence or absence of rainfall during the calling period, and lunar stage (Table 1). Danum Valley Conservation Area operates a weather station that has been collecting weather data continuously since 1985. The weather data used in the present study were accessed from the South East Asia Rainforest Research Partnership (SEARRP) website (searrp.org). Lunar stage was included as a categorical variable of either 'new', 'waxing', 'full' or 'waning' and was obtained using the 'lunar' package [63] in the R programming environment [64]. We also included a categorical variable (which we termed 'calling period') that indicated whether the calling event was in the night (00:00–06:00LT), morning (6:00–12:00LT), afternoon (12:00–18:00LT) or evening (18:00–00:00LT). We were able to identify rain events from the LTSAs and based on this we included a binary variable indicating whether rain occurred during a particular calling period.

For our first analysis we used a binary variable (presence or absence of argus calling event) as the outcome in our model. As we modeled a binary outcome, we fit the models using a binomial distribution. We created a series of six generalized linear mixed models using the package 'glmmTMB' [65] with argus calling events as the binary outcome, along with calling period and either daily maximum temperature (˚C), daily minimum temperature (˚C), rain during the calling period (binary), rainfall the previous day (mm/24h), or rainfall on the current day

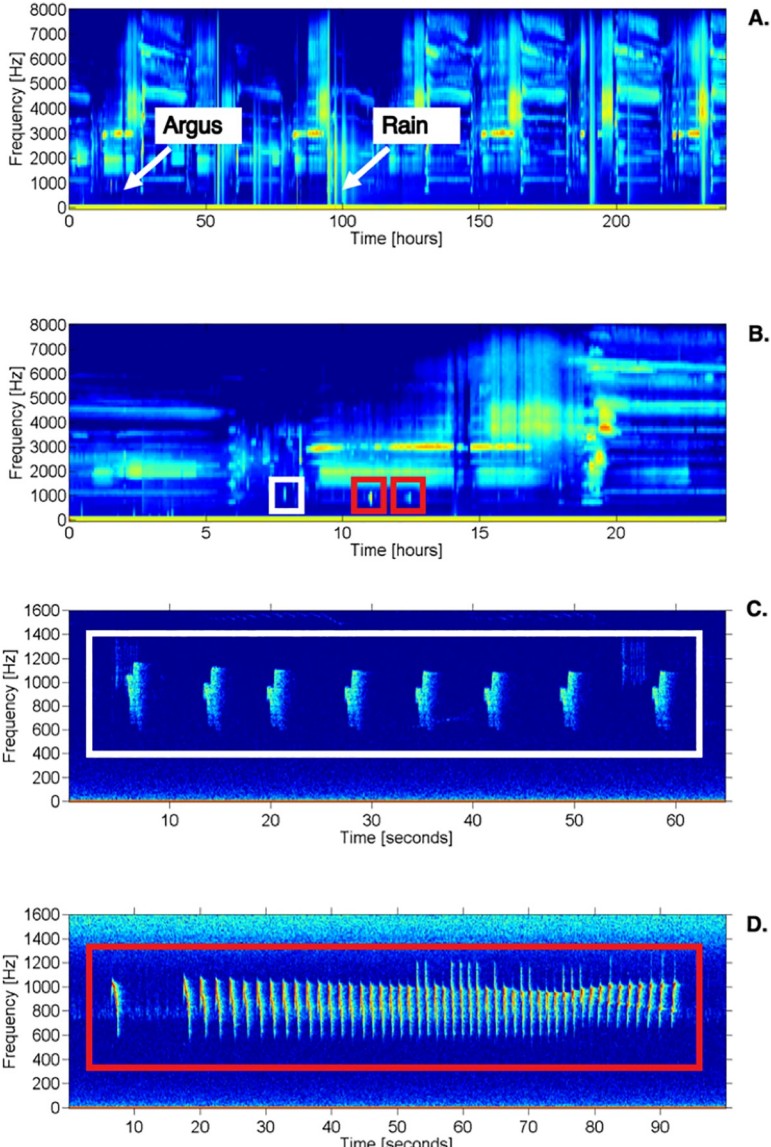

**Fig 2.** Representative long-term spectral average plots (LTSAs) of a 10-day period (A), and a 24-hour period (B) in Danum Valley Conservation Area, Sabah, Malaysia, along with representative spectrograms of a great argus short call (C) and long call (D). A) Arrows indicate presence of great argus calling events and rain. B) The white box on the left indicates a short call type and the two red boxes on the right indicate long call types. See main text for description of LTSA settings. C-D) Spectrograms were made using Triton software with a 2400-point (150 ms) FFT window size, Hann window, and 85% overlap. Spectrogram equalization was turned off.

(mm/24h) as predictors (see Table 1 for description). We also included a null model that did not include any of the environmental predictors. Each unique model represented a specific hypothesis (*e.g.*, great argus have a lower likelihood of calling when the minimum overnight temperature is lower) and all models included recorder and date as random effects.

For our next analysis, we included total number of great argus calls during a particular calling period as our outcome variable. Representative spectrograms of great argus long and short calls are provided in Fig 2C and 2D; the counts in our models indicate the number of distinct calls as shown in the spectrograms. For these models, as we were modeling count data, we

**Table 1. Summary of outcome, predictor variables and random effects included in models of great argus calling.**

| Outcome variable | Description | Total calls | Range of number of calls in calling period |
|---|---|---|---|
| *Argus calling event* | A binary variable indicating presence of argus calls during a calling period. | ~ | |
| *Number of argus calls (all)* | Number of argus calls during all calling periods. | 2,738 | 0–48 |
| *Number of argus calls (18:00–06:00)* | Number of argus calls during the evening and early morning periods. | 466 | 0–16 |
| **Predictor variable** | | **Mean and SD** | **Range** |
| *Maximum Temperature* | Daily maximum temperature (˚C) | 31.4 ± 1.4 | 25.4–34.3 |
| *Minimum Temperature* | Daily minimum temperature (˚C) | 23.4 ± 0.7 | 22.2–25.4 |
| *Rain during calling period* | A binary variable indicating whether rain occurred during the calling period | ~ | ~ |
| *Rainfall previous day* | Total precipitation (mm/24h) taken at 08:00 each day. | 6.9 ± 7.1 | 0.0–31.7 |
| *Rainfall current day* | Total precipitation (mm/24h) taken at 08:00 each day. | 6.9 ± 7.1 | 0.0–31.7 |
| *Calling period* | A categorical variable indicating whether the calling event was in the early morning (00:00–06:00), morning (6:00–12:00), afternoon (12:00–18:00) or evening (18:00–23:59) | ~ | ~ |
| *Call type* | A categorical variable indicating whether calls in a particular calling period were long calls or short calls. | ~ | ~ |
| *Lunar cycle* | Categorical variable indicating whether the moon was full, new, waxing or waning. | ~ | ~ |
| **Random effects** | | | |
| *Recorder* | Recorder on which the calling event occurred. | ~ | ~ |
| *Date* | Date on which the calling event occurred. | ~ | ~ |

used a negative binomial distribution with a log-link function. To account for differences in recording duration (*i.e.*, sampling effort) we included the number of recorders that were recording on a particular day (log-transformed) as an offset in each model [65]. As described above, we created a series of six generalized linear mixed models in the 'glmmTMB' package. These models also included a random effect for recorder location and date.

**Analyses on subset of calls which only occurred from 18:00–06:00LT.** As previous reports classified great argus as 'strictly diurnal' [55] we were also interested to see if calling events during between 18:00–06:00LT were influenced by different environmental factors (*e.g.*, lunar cycle) than calling during the entire 24-hour calling period, so we ran both of the analyses described above on a subset of the data which only included calling events between 18:00–06:00LT. As above, we included either daily maximum temperature (˚C), daily minimum temperature (˚C), rain during the calling period (binary), rainfall the previous day (mm/24h), or rainfall on the current day (mm/24h) as predictors. In addition, we added one model that included lunar cycle as a predictor and another that included both lunar cycle and presence of rain during the calling period as predictors. We did not include calling period as a predictor in these models. We modeled either presence or absence of calls or number of calls between the hours of 18:00–06:00LT as outlined above and included a null model that did not include any of the environmental predictors.

**Modeling usage of distinct call types.** As our dataset consisted of two different call types (short and long call), we wanted to test for differences in the use of the two call types across a 24-hour period. For this analysis we focused only on calling periods in which there were great argus calls. We included the environmental predictors outlined above, as well as call type (short call or long call) as a predictor and number of calls as the outcome. We were modeling count data, so we used a negative binomial distribution and included log-transformed number of recorders as an offset. We added an interaction between calling period and call type to this

set of models to test if differences in call use were dependent on calling period. We compared these models to a null model that only included calling period but none of the other predictors.

**Call and response.** A portion of the calls in our dataset appeared to be elicited in a call and response, and we were interested to see if call and response was more likely depending on the call type. Although it would be possible in the presence of a human observer to determine the directionality of calls, we had to rely on passive acoustic data only, so we conservatively defined a call as a 'response' if the start time of the call occurred during the time the initial call was recorded (*e.g.* if there was temporal overlap). Temporal overlap in calls provides strong evidence that the calls must have come from more than one individual. Therefore, for each call in our dataset we included a binary variable indicating whether there was a call elicited in response. We created a series of two generalized linear mixed models in the 'glmmTMB' package with a binary outcome indicating whether a particular call elicited a response. The first model included a null model that only had a random effect for recorder location and the second model included call type as a predictor.

**Model comparison using Akaike information criterion.** For each set of analyses we compared all candidate models using Akaike information criterion (AIC) adjusted for small sample sizes (AICc) using the 'bbmle' R package [66]. We tested for multicollinearity of predictors in our top models by using the 'check_collinearity' function in the 'performance' package [67] and we used the 'DHARMa' package to test for normality of residuals [68]. To provide an estimate of how well our top models fit the data, we calculated a pseudo-$R^2$ value using the 'MuMIn' package [69]. All analyses were done in the R programming environment [64].

## Permits

The research presented here adhered to all local and international laws. Institutional approval was provided by Cornell University (IACUC 2017–0098). Sabah Biodiversity Centre and the Danum Valley Management Committee provided permission to conduct the research under permit number JKM/MBS.1000-2/2 JLD.9 (62).

## Results

We report the results of an analysis of 1,618 short calls and 1,120 long calls (n = 2,738 total calls) from 14568 hours of autonomous recordings in Danum Valley Conservation Area, Sabah, Malaysia (Table 2). The frequency range of great argus calls was between 600–1,600 Hz (Fig 2). Long calls were 82 s in duration on average (range = 5–360 s) and the short calls were 2 s in duration on average (range = 1–380 s). We found that argus called throughout the 24-hour period but were more likely to call in the morning between 06:00–12:00LT (Fig 3) and that there was substantial spatial variation in calling events at across the array (Table 2).

### Environmental predictors of great argus calling events

We ran two separate analyses to investigate the environmental predictors of great argus calling events. The first analysis included a binary outcome variable indicating whether there was a great argus calling event during that calling period or not, and the second analysis included the total number of great argus calls during a particular calling period. For our first set of models, we found that great argus calling events (as a binary outcome) were more likely to occur during the morning calling periods compared to other calling periods, and least likely to call during the evening calling period, although calls were detected during all times of the day. We also found that argus calling events were less likely if there was rain during a particular calling

**Table 2. Total number of recording hours, days and number of great argus calling events summarized by autonomous recorder.**

| Recorder | Total Days | Total Hours | Total Calls | Short Calls | Long Calls |
|---|---|---|---|---|---|
| S10 | 22 | 516 | 45 | 6 | 39 |
| S11 | 61 | 1464 | 673 | 448 | 225 |
| S12 | 66 | 1584 | 111 | 2 | 109 |
| S13 | 49 | 1176 | 52 | 8 | 44 |
| S14 | 32 | 768 | 304 | 245 | 59 |
| S15 | 64 | 1536 | 13 | ~ | 13 |
| S16 | 54 | 1296 | 106 | 52 | 54 |
| S17 | 80 | 1920 | 406 | 209 | 197 |
| S18 | 48 | 1164 | 673 | 466 | 207 |
| S19 | 98 | 2352 | 181 | 17 | 164 |
| S20 | 33 | 792 | 174 | 165 | 9 |
| TOTALS | 607 | 14568 | 2738 | 1618 | 1120 |

Recorder S10 was deployed later in the field season, which is why it has fewer recording days, and the variable recording days and hours for the remaining recorders are due to differences in battery-life and/or unit malfunctioning.

period (Table 3 and Fig 4). Our top model comprised 82% of the model weight when doing model comparison with AIC, and included calling period and presence of rain during the calling period as reliable predictors. The top model performed better than the intercept only model ($\Delta AIC = 5.5$; <0.01% of model weight), and the pseudo-$R^2$ value indicated that the

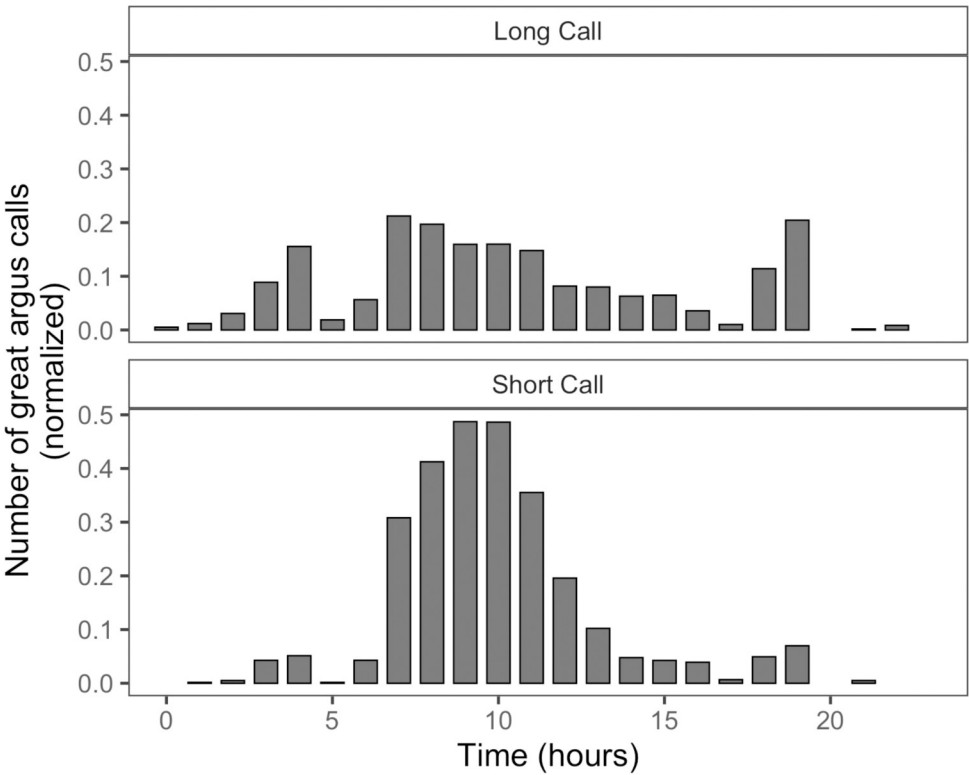

**Fig 3. Number of great argus calling events (n = 2,738) normalized by total recording time for each hour in Danum Valley Conservation Area.** Each time bin represents one hour on the 24-hour clock (local time).

**Table 3. Regression coefficients for great argus calling events modeled as either a binary response (presence or absence of great argus calls) or the total number of calls during a calling period.**

|  | Binary Model Top | Binary Model Intercept | Count Model Top | Count Model Intercept |
|---|---|---|---|---|
| Morning Calling Period | 0.67 ** | 0.72 ** | 1.12 ** | 1.19 ** |
|  | [0.46, 0.88] | [0.51, 0.93] | [0.92, 1.31] | [0.99, 1.38] |
| Night Calling Period | -0.26 ** | -0.19 | -0.28 ** | -0.19 |
|  | [-0.51, -0.02] | [-0.43, 0.04] | [-0.51, -0.05] | [-0.42, 0.04] |
| Evening Calling Period | -0.56 ** | -0.52 ** | -0.68 ** | -0.60 ** |
|  | [-0.82, -0.31] | [-0.77, -0.26] | [-0.94, -0.42] | [-0.86, -0.35] |
| Rain (Binary) | -0.25 ** |  | -0.35 ** |  |
|  | [-0.42, -0.07] |  | [-0.51, -0.19] |  |
| logLik | -1824.66 | -1828.41 | -3381.56 | -3390.78 |
| AIC | 3663.31 | 3668.82 | 6779.13 | 6795.56 |
| BIC | 3706.16 | 3705.55 | 6828.10 | 6838.41 |

The table includes a summary of the highest-ranked model based on AIC comparison along with the intercept only model which included only calling period as a predictor. We considered predictors which are indicated with ** as reliable predictors of great argus calling, as the 95% confidence intervals did not overlap zero.

predictor variables explained ~6% of the variance and the entire model (predictors and random effects) explained ~21% of the variance.

The top model for our second set of models–which included the total number of great argus calls during a calling period as the outcome–included morning calling period as positive predictors of number of great argus calls, whereas evening and night calling periods, along

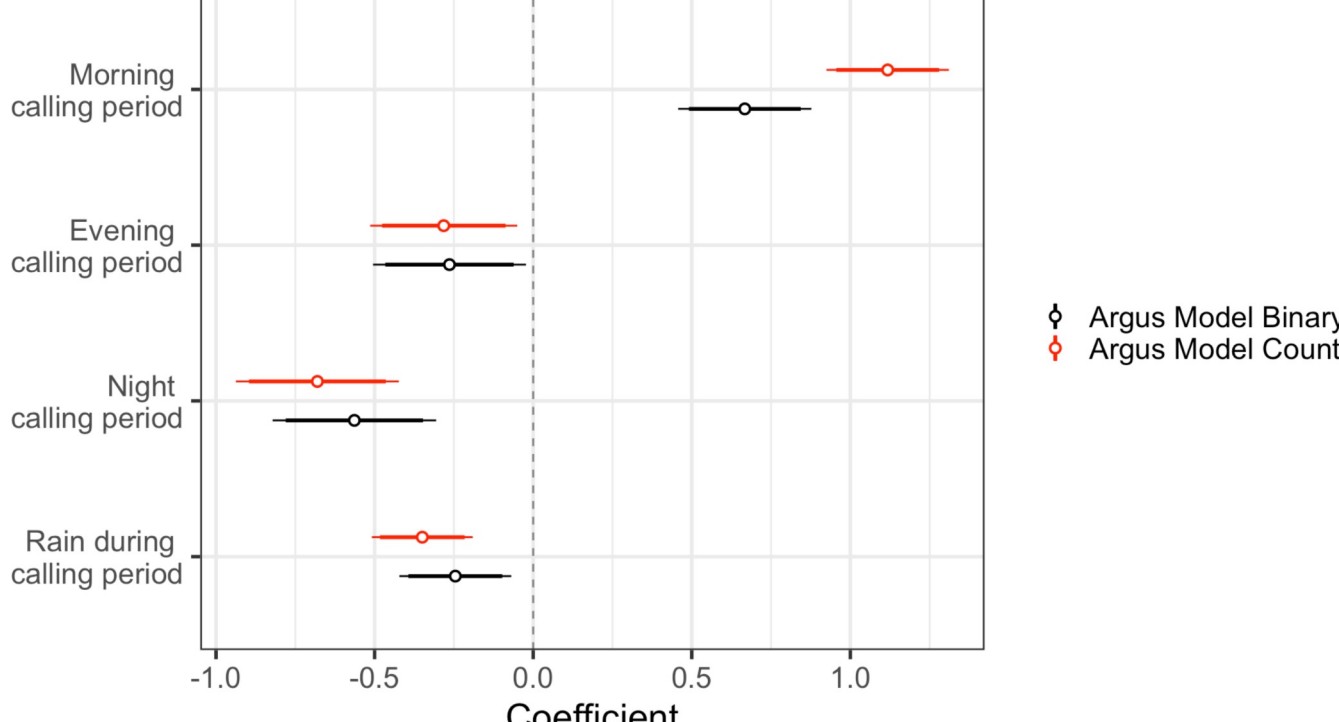

**Fig 4. Coefficient estimates for top models predicting argus calls (either presence/absence or total number of calls) during a calling period.** Argus were more likely to call during the morning calling period, and they were less likely to call if there was rain during a calling period. We considered predictors reliable if the 95% confidence intervals did not overlap zero.

**Table 4. Regression coefficients for models of great argus calling events which occurred from 18:00–06:00 modeled as either a binary response (presence or absence of great argus call) or the total number of calls.**

|  | Binary Model Top | Count Model Top |
|---|---|---|
| Waning Moon | 0.08 | -0.06 |
|  | [-0.45, 0.62] | [-0.45, 0.32] |
| New Moon | 0.56 ** | 0.39 ** |
|  | [0.05, 1.07] | [0.05, 0.73] |
| Waxing Moon | 0.32 | 0.18 |
|  | [-0.20, 0.83] | [-0.19, 0.54] |
| Rain (binary) | -0.94 ** | -1.00 ** |
|  | [-1.34, -0.54] | [-1.29, -0.72] |
| logLik | -378.21 | -698.27 |
| AIC | 770.41 | 1412.54 |
| BIC | 801.90 | 1448.65 |

We considered predictors which are indicated with ** to be reliable as the 95% confidence intervals did not overlap zero.

with the presence of rain during a calling period, were negative predictors (Table 3 and Fig 4). Our top model for total number of great argus calls performed substantially better than the intercept only model (ΔAIC = 16.4; <0.001% of model weight) and comprised 99% of the model weight. Based on the pseudo-$R^2$ value the predictor variables explained ~18% of the variance and the entire model (predictors and random effects) explained ~50% of the variance.

## Analyses on subset of calls which only occurred from 18:00–06:00LT

We re-ran our analyses on a subset of calls which occurred during the nighttime hours (n = 466 calls) and modeled the outcome as both presence/absence and total count of argus calls during a calling period. We found that when modeling a binary outcome, great argus were more likely to call during new moon periods, and were less likely to call when there was rain during the calling period (Table 4 and Fig 5). Our top model comprised 100% of the model weight and performed better than the intercept only model (ΔAIC = 24.8; <0.001% of model weight), and the pseudo-$R^2$ value indicated that the predictors explained ~6% of the variance and the entire model (predictors and random effects) explained ~ 30% of the variance. When modeling the total number of argus calls we again found that moon phase (new) was a positive predictor of total number of calls, and presence of rain was a negative predictor (Table 4 and Fig 5). The top model comprised 100% of the model weight and performed better than the intercept only model (ΔAIC = 60.3; <0.001% of model weight), and the pseudo-$R^2$ value indicated that the predictors explained ~15% of the variance and the entire model (predictors and random effects) explained ~77% of the variance.

## Modeling usage of distinct call types

We were interested to see if there were population-level differences in the use of the two call types (short call and long call). To test this, we created a series of models with number of argus calls in a calling period as the outcome. In addition to the environmental variables outlined above we as included a predictor variable indicating whether the calls in the calling period were long calls or short calls. Our top model for this analysis did not include any environmental predictors, but it did include an effect of morning calling period, and a positive interaction between short calls and morning calling periods, indicating that great argus called more during

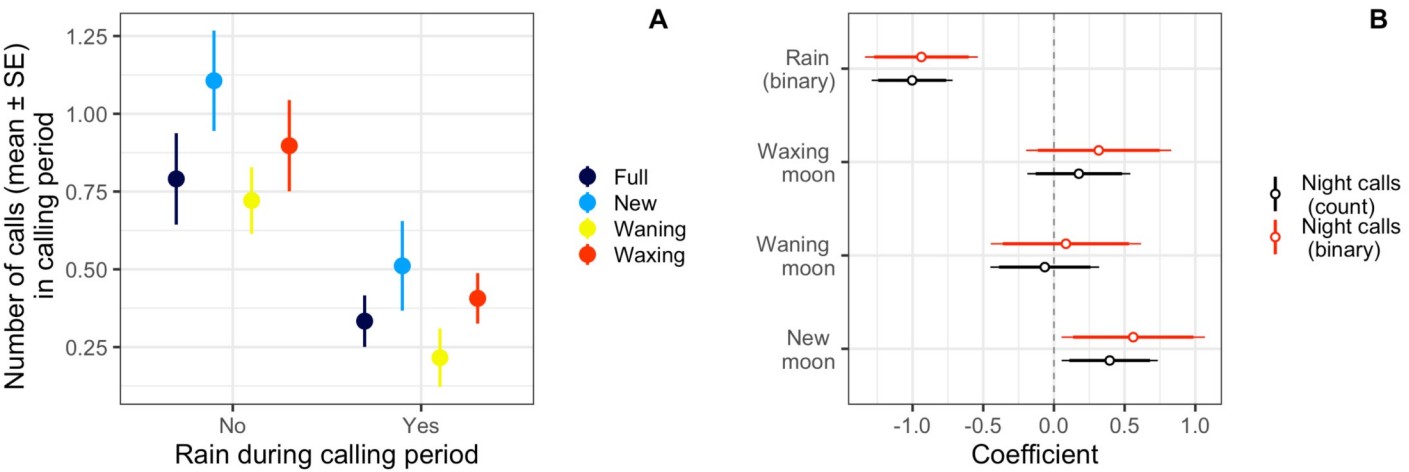

**Fig 5.** Number of calls (mean ± SE) in calling period as a function of rain and lunar cycle (A) and top model coefficient plots (B) for great argus calling events which occurred from 18:00–06:00. For the coefficient plots (B) calls were modeled as either a binary response (presence or absence of great argus call) or the total number of calls. When modeling both the binary and continuous outcome, we found that great argus were more likely to call if there was a new moon, and less likely to call if there was rain during the calling period. We considered predictors reliable if the 95% confidence intervals did not overlap zero.

the morning calling period, and that calls during the morning calling period were more likely to be short calls (Fig 6). The top model accounted for 75% of the model weight and performed substantially better than the intercept only model which only contained calling period as a

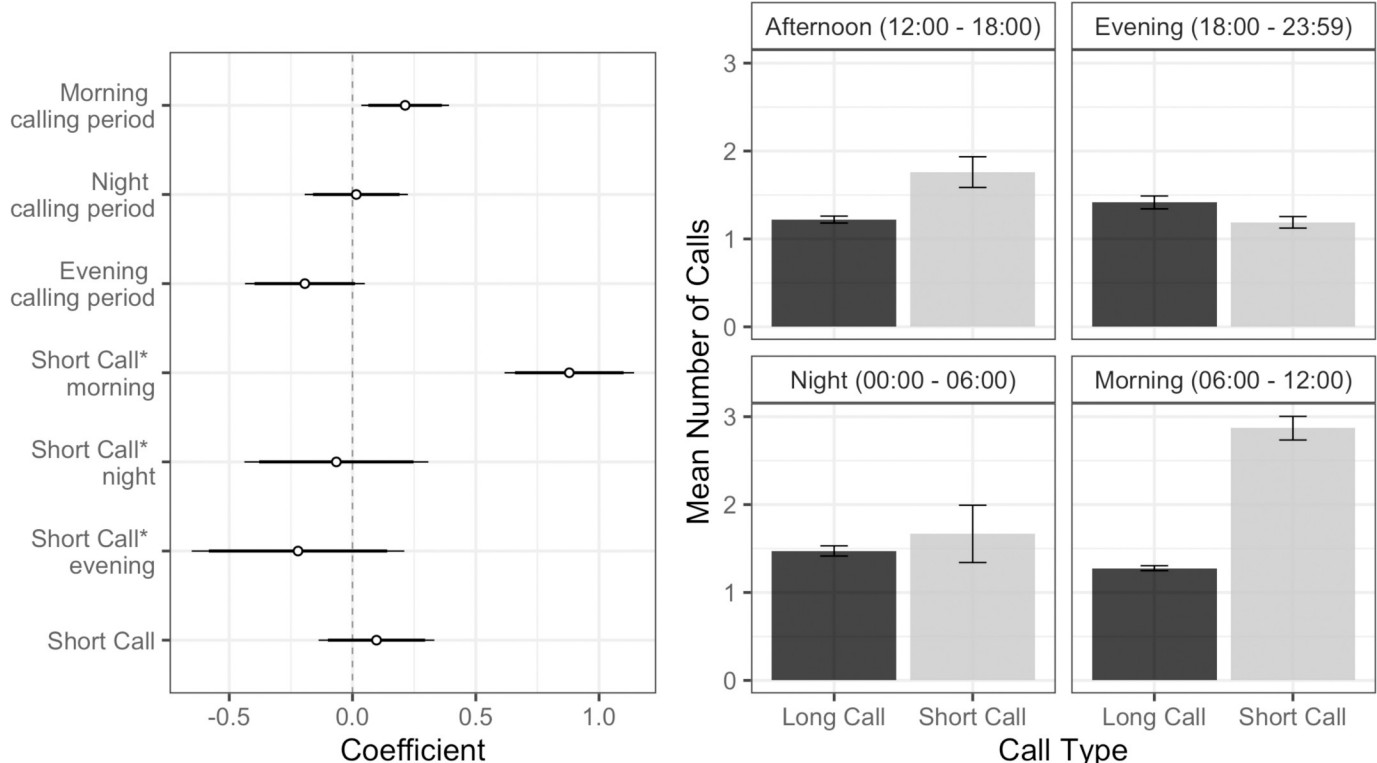

**Fig 6. Coefficient plot for the top model of great argus use of different call types (L) and mean number ± standard error for each calling period (R).** Great argus called more often during morning calling periods, and when they called in the morning the tended to emit short calls.

**Table 5. Akaike's information criterion (AICc) model comparison results for the five separate analyses investigating argus calling behavior in Danum Valley Conservation Area, showing the top two models and the null model for each analysis.**

| Models | AICc | Δ AICc | DF | Weight |
|---|---|---|---|---|
| **Presence/absence of argus calls ~** | | | | |
| Rain (binary) + calling period | 3663.34 | 0.00 | 7 | 0.82 |
| Intercept | 3668.85 | 5.50 | 6 | 0.05 |
| Rain previous day | 3668.88 | 5.53 | 7 | 0.05 |
| **Number of argus calls ~** | | | | |
| Rain (binary) + calling period | 6779.17 | 0.00 | 8 | 0.99 |
| Temperature (max) + calling period | 6788.43 | 9.26 | 8 | 0.01 |
| Intercept | 6795.59 | 16.42 | 7 | 0.00 |
| **Presence/absence of argus calls (18:00–06:00LT) ~** | | | | |
| Rain (binary) + lunar phase | 770.58 | 0.00 | 7 | 1.00 |
| Minimum temperature + lunar phase | 785.05 | 14.47 | 7 | 0.00 |
| Intercept | 796.92 | 26.34 | 4 | 0.00 |
| **Number of argus calls (18:00–06:00LT) ~** | | | | |
| Rain (binary) + lunar phase | 1412.76 | 0.00 | 8 | 1.00 |
| Minimum temperature + lunar phase | 1451.28 | 38.53 | 8 | 0.00 |
| Intercept | 1473.03 | 60.27 | 4 | 0.00 |
| **Number of argus calls ~** | | | | |
| Call type * calling period | 3795.16 | 0.00 | 10 | 1.00 |
| Rain (binary) + lunar phase | 3991.47 | 196.32 | 11 | 0.00 |
| Intercept | 4069.73 | 274.57 | 7 | 0.00 |

We modeled the number and presence or absence of argus calls during a calling period, the number and presence or absence of argus calls during the night (18:00–06:00LT) and the number of calls as a function of call type.

predictor (ΔAIC = 274.5; <0.001% of model weight). The pseudo-R2 value indicated that the predictors in our top model explained 37% of the variance and the entire model explained 72% of the variance. A summary of AICc model comparison for the top two models and the null model from all analyses done in the current study is shown in Table 5.

## Call and response

Approximately 6% of the calling events were emitted in a call and response, wherein the start of one call happened before the completion of an earlier call, indicating that the calls must have come from two or more individuals. We found that long calls were much more likely to elicit a response than a short call (estimate = 1.6096; SE = 0.24), and the model containing call type as a predictor substantially outperformed the null model (ΔAIC = 48.0; <0.001% of model weight).

## Discussion

Here we provide one of the first descriptions of the vocal behavior of great argus pheasants in Malaysian Borneo. We found that although great argus called more often during the morning calling period (06:00–12:00LT) they did not limit their calling to this time, and calling events occurred throughout the day. We found that a substantial portion (~20%) of their calls were emitted at night between the hours of 18:00–06:00LT. Presence of rain (as detected via aural and visual inspection of the LTSAs) during a calling period reduced the likelihood of a great argus calling. And for nighttime calling periods, great argus tended to call more at night if

there was a new moon, and less when there was rain during the calling period. We found that both call types were emitted throughout the day, but short calls were more common during the morning calling period. As very little is known about the behavioral ecology of great argus on Borneo it is unclear if the patterns of calling behavior we documented are typical, or if there was increased vocal output because our study happened to occur during their breeding season. What is clear is that similar to other studies on vocal animals, great argus calling behavior is influenced by extrinsic factors including rain, temperature, and moonlight. Future, long-term studies of great argus vocal behavior across years (and including during mast fruiting [70] periods) will help further our understanding of how environmental and ecological factors influence their vocal behavior.

One of the few descriptions of great argus call types (from peninsular Malaysia [50]) noted that long calls were given from any point in the forest, and are considered to be territorial in nature, whereas the short calls were emitted more often during the morning from the male dancing grounds, and repeated in a way that may facilitate localization by female great argus [50]. The patterns of call timing in great argus at our study site are in alignment with these observations, as we found that short calls were more likely to be emitted in the morning than long calls, and that the frequency of short calls decreased over the 24-hour period. In addition, the fact that long calls also elicited more responses than short calls provide further evidence of their territorial function. Our results were also similar to reports from East Kalimantan, Indonesia which report that long calls occur less frequently [56]. We also noted a peak in vocal activity of long calls between 18:00–19:00LT. Many diurnal birds engage in a dusk chorus in addition to a dawn chorus but the function of the dusk chorus is still a topic of debate [12, 71], and the reason for increased vocal activity in great argus around dusk is not clear.

Great argus on Sumatra were classified as being 'strictly diurnal' based on the analysis of 943 photographs taken using camera traps deployed between 1998–2001 [55]. At our site, approximately 20% of the recorded 2,738 great argus calls were emitted between the hours of 18:00–06:00LT, indicating that great argus are oftentimes vocally active at night. The context under which the great argus call at night is unclear. Our study is not the first to report nighttime calling activity in great argus (see [50]), but is in contrast to previous reports based on data collected using camera traps. It may be that great argus do not move from their sleeping site when they vocalize at night, which would lead to discrepancies between camera trap and acoustic data in documented activity patterns. It is also possible that differences in our results compared to great argus on Sumatra may also be related to differences in predation pressure. Differences in terrestriality between Bornean and Sumatran orangutans have been attributed to the presence of tigers on Sumatra [72, 73], and it is possible that differences in night time activity patterns between great argus on Borneo and Sumatra have also been shaped by differences in predation pressure.

Another line of evidence that nighttime calling may be influenced by predation pressure is the fact that we found that great argus called more often during new moon periods. Lunar phobia is a common phenomenon wherein animals reduce their activity patterns during periods of high moonlight [74]. A meta-analysis on bats revealed that there was a significant negative relationship between bat activity and moonlight intensity [75]. It is possible that lunar phobia in bats is the result of increased predation pressure during high moonlight intensity, or it may be related to activity patterns of their prey, as katydids and their bat predators were both shown to have increased activity patterns during periods of low moon intensity [76]. Over 30 species of birds have been shown to increase vocal output during full moon periods or in artificial light environments (reviewed in [12]), whereas relatively few have been shown to have the opposite pattern. Leach's storm petrels showed a decrease in nocturnal activity during high moonlight periods, which was attributed to increased predation risk during high moonlight

periods [77]. Ovenbirds and white-throated sparrows also showed a decrease in vocal output during full moon periods, which the authors attribute to potential increased predation risk of calling during full moon periods [78]. It is also possible that because there is less available moonlight during new moon periods great argus rely less on visual displays and more on vocal communication.

Our results add to the growing body of literature indicating that abiotic factors can shape the vocal behavior of birds. For example, in temperate birds the timing of the dawn chorus was shown to vary with lunar phase, temperature, cloud cover, and precipitation, providing evidence that extrinsic abiotic factors can have an influence on call timing [79]. Although changes in temperature and day length are less pronounced in the tropics, birds in equatorial lowland Amazonas were shown to modify the start of the dawn chorus in response to slight changes in day length [80]. We found that rain during a particular calling period lead to a decrease in great argus calling, which could be related to changes in energy expenditure.

In the present study, we use PAM to provide important insight into the behavioral ecology of the cryptic yet vocal great argus, but there is still much more to be learned. As mentioned above, little is known about great argus reproductive behavior on Borneo, so it is hard to know if our results are typical, or whether vocal activity patterns would change if we conducted our study during a different time of the year. We found differences in use of the different call types which were consistent with the proposed functions territorial versus female attraction [50] but the use of playbacks (*e.g.*, [81]) could help further elucidate differences in function of the two call types. In addition, the differences in reported activity patterns—i.e. 'strictly diurnal' in Sumatra [55] versus vocally active at night (this study; [50])—may be related to differences in the ecology or differences in study design. A particularly useful future avenue of research for improving our understanding of great argus behavior will be the use of paired camera traps and acoustic recorders.

The forests of Southeast Asia are undergoing some of the fastest rates of deforestation in the world [82]. Here, we show how PAM can be used to monitor spatial and temporal distribution of calling behavior of great argus pheasants which are extremely difficult to study using human observers. PAM approaches have important conservation implications, particularly if they can be used to improve understanding of how great argus respond to anthropogenic disturbance. There has been increasing interest in applying PAM approaches to estimate occurrence and density of vocal animals in terrestrial environments [83–86]. A crucial next step in PAM of great argus pheasants will be in the development of effective methods of estimating occurrence and density across large spatial scales. Future monitoring approaches which allow for rapid assessment will be critical for effective management and conservation of great argus pheasants across their range.

## Acknowledgments

We thank Yoel Majikil for his assistance with data collection for this project. We also thank Ashakur Rahaman for his assistance with making the map. We also gratefully acknowledge Lynn Marie Johnson at the Cornell University Statistical Consulting Unit for statistical advice.

## Author Contributions

**Conceptualization:** Dena J. Clink, Tom Groves, Abdul Hamid Ahmad, Holger Klinck.

**Data curation:** Dena J. Clink, Tom Groves.

**Formal analysis:** Dena J. Clink, Tom Groves.

**Funding acquisition:** Dena J. Clink.

**Investigation:** Dena J. Clink.

**Methodology:** Dena J. Clink, Holger Klinck.

**Project administration:** Dena J. Clink.

**Resources:** Abdul Hamid Ahmad, Holger Klinck.

**Validation:** Dena J. Clink, Tom Groves.

**Visualization:** Dena J. Clink, Tom Groves, Holger Klinck.

**Writing – original draft:** Dena J. Clink, Tom Groves, Abdul Hamid Ahmad, Holger Klinck.

**Writing – review & editing:** Dena J. Clink, Tom Groves, Abdul Hamid Ahmad, Holger Klinck.

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
