## [Decision Letter · Decision Letter 0]

28 Oct 2020

PONE-D-20-28926

Not by the light of the moon: investigating circadian rhythms and ecological predictors of calling in Bornean great argus

PLOS ONE

Dear Dr. Clink,

Thank you for submitting your manuscript to PLOS ONE. After careful consideration, we feel that it has merit but does not fully meet PLOS ONE’s publication criteria as it currently stands. Therefore, we invite you to submit a revised version of the manuscript that addresses the points raised during the review process.

We look forward to receiving your revised manuscript.

Kind regards,

Dennis M. Higgs

Academic Editor

PLOS ONE

Journal Requirements:

3.We note that [Figure(s) 1] in your submission contain [map/satellite] images which may be copyrighted. All PLOS content is published under the Creative Commons Attribution License (CC BY 4.0), which means that the manuscript, images, and Supporting Information files will be freely available online, and any third party is permitted to access, download, copy, distribute, and use these materials in any way, even commercially, with proper attribution. For these reasons, we cannot publish previously copyrighted maps or satellite images created using proprietary data, such as Google software (Google Maps, Street View, and Earth). For more information, see our copyright guidelines: http://journals.plos.org/plosone/s/licenses-and-copyright.

1.    You may seek permission from the original copyright holder of Figure(s) [1] to publish the content specifically under the CC BY 4.0 license. 

Additional Editor Comments (if provided):

As you can see both reviewers were highly complementary of your work. Reviewer #2 does ask for some clarification throughout the manuscript so please review those suggestions carefully. In a few places they suggest additional statistical analyses so please consider those but only use them if the authors feel it would be central to the goals of the research and would add clarity to the outcomes.

Reviewers' comments:

Reviewer's Responses to Questions

**Comments to the Author**

1. Is the manuscript technically sound, and do the data support the conclusions?

Reviewer #1: Yes

Reviewer #2: Yes

2. Has the statistical analysis been performed appropriately and rigorously? 

Reviewer #1: Yes

Reviewer #2: Yes

3. Have the authors made all data underlying the findings in their manuscript fully available?

Reviewer #1: Yes

Reviewer #2: Yes

4. Is the manuscript presented in an intelligible fashion and written in standard English?

Reviewer #1: Yes

Reviewer #2: Yes

5. Review Comments to the Author

Reviewer #1: I reviewed this paper before for Ibis and unfortunately they did not accept it - clearly the different reviewers and the editor disagreed. I thought then, and I think now, that this is a sound paper and the changes I suggested when it was under review for Ibis have been incorporated so I have nothing else to suggest.

Just fix the referencing style on page 5 (names of the authors rather than numbers).

Vincent Nijman, Oxford, 8 October 2020

Reviewer #2: REVIEW OF MANUSCRIPT PONE-D-20-28926

This is a well-written manuscript describing the vocal activity of the Bornean great argus, a threatened species that is particularly known for its amazing visual display to attract females. The methods and analyses are adequate, and the findings are relevant for understanding more about the species’ behavior and ecology. Should be an important contribution to the field.

I have made some minor commentaries throughout the manuscript regarding some points that were unclear for me. The main issue I’d like to highlight would be to improve clarity in the statistical methods used and to consider making some adjustments that are detailed below. Finally, I’d like to commend the authors for the nice work done here.

Line-by-line commentaries

Title

When I read “ecological predictors”, I expected to find goals associated with population or interactions, as competition. But the predictors used could be names as environmental variables. Perhaps you could change to “…: environmental drivers of circadian calling activity…”.

Introduction

L. 43-51: The opening paragraph lacks a message of what’s coming next. It highlights the prospects and research gaps of PAM, but that’s just the method. It would be more interesting to be introduced with a message related to the core ecological aspect of the manuscript. For instance, why is vocal behavior important and what are the drivers of calling activity? Then, in the next paragraphs, you could describe why we still know little about these behaviors, and how PAM can help to fill this gap.

L. 47: There are large-scale and temporal efforts to obtain calling behavior data, such as citizen science projects. Perhaps you could say that it requires substantial efforts, not that it is impossible.

L. 97-102: It would be interesting to see a paragraph introducing the potential effects of the lunar cycle in calling activity, in general, and also for birds. Additionally, it would also be interesting if the function of these calls could be introduced. Are they courtship calls? How are they characterized? Are there differences in the function of these calls?

Methods

L. 107: Please, include the unit type of the coordinates (degrees?), indicate latitude and longitude, and datum (WGS84?).

L. 113: Were the recordings made in stereo?

L. 116: Include a few words describing how are gibbon calls (frequency range, duration, stereotyped, or not).

L. 145-146: All the monitored hours were inspected by the observer?

L. 147: Was the 10-day period used in the analysis?

L. 151: It is really difficult to see the differences between the two call types in the LTSA. Did you also zoomed in the spectrogram to distinguish between the call types, or used always 24-hour LTSA + aural discrimination? What are the advantages of using LTSA and “zoomed-in” spectrograms, like those available in Audacity/Raven? This information can be useful for the reader interested in using LTSA.

Figure 2: The resolution from the downloaded figure is not good. For instance, in figure 2.A, it is not clear what the first arrow points to, although the legend says it is a call. Particularly, if this is the resolution used to identify events of call by one observer, it raises doubts if it was possible to discriminate calls. Please, indicate in the methods if the observer used the entire range of 0-8000 Hz to screen for the calls, or the 0-1600 Hz (shown in 2.C and 2.D).

L. 154: Please, add full LTSA meaning.

L. 159: Add “software” after Triton.

L. 179: You could add the models and its variables in table 1 in the form of “questions” or “set” of tests. For instance,

1. 24-hour calling activity?

1.1 Presence/absence of call events ~ variables

1.2. Total number of calls ~ variables

2. Night activity?

2.1 Pres/absence of call events (18:00-06:00) ~ variables

2.2 Total number of calls (18:00-06:00) ~ variables

3. Differences in call type?

Number of calls ~call type * calling period

L. 181: Each model contains only one predictor? If so, please amend this to this sentence.

L. 187: Are the number of argus calls the number of notes, or call events? Please, describe this in the previous section.

L. 209: Did you also included a model with intercept only (null model)? This should be done to understand how different your best model is from a null one.

L. 221: Also add the total number of hours/days evaluated.

L. 223: Did you measured the lengths from all calls or a subset of them?

L. 221-224: If the previous sentence is “yes”, another information that would be interesting to improve our understanding of the ecology of the great argus would be to understand if any of the meteorological variables and period of the day influence the length of calls. I know this is not included in the goal but consider doing it since you already got the data.

L. 225: You could also test if a “response” was more frequent in some period of the day or for some call type. Depending on the results, it may indicate that it can function better to male-male interactions, for instance.

Results

L. 250: Please, include the hours corresponding to the morning and evening periods.

L. 253: There is no information about a model representing the effect of rain in table 3.

L. 255: Here you say the model was compared with a null one, but also include that in the methods. Additionally, it is not clear if you ranked each model with a null one, or if you ranked all models + null. I think the last option is better, as we can check if there are models equally plausible.

L. 256: It would be nice to see a table with the models ranked, containing delta AICc, AICc, and the weights.

L. 304: Previously, I had understood that you used presence/absence from the night period as the response variable. But why the calling period enter as a predictor here?

Discussion

L. 348: You could explore if there are differences in calling activity between months, and indicate, for instance, which month were the individuals engaging more in calling activity. These suggestions for new analysis help in filling the gap that is raised in this sentence (little is known about the behavioral ecology of great argus…).

L. 378-379: Would it also be possible that visual display is unfavored during new moon, and thus engaging in acoustic advertising would be a strategy of communication? Check if the other references indicate a role of moon phase in the pattern of visual display.

6. PLOS authors have the option to publish the peer review history of their article (what does this mean?). If published, this will include your full peer review and any attached files.

Reviewer #1: No

Reviewer #2: No

---

## [Author Response · Author response to Decision Letter 0]

8 Jan 2021

Please see attached response to reviewers document.

---

## [Editor Report · Decision Letter 1]

22 Jan 2021

Not by the light of the moon: investigating circadian rhythms and environmental predictors of calling in Bornean great argus

PONE-D-20-28926R1

Dear Dr. Clink,

We’re pleased to inform you that your manuscript has been judged scientifically suitable for publication and will be formally accepted for publication once it meets all outstanding technical requirements.

Kind regards,

Dennis M. Higgs

Academic Editor

PLOS ONE
---

## [Editor Report · Acceptance letter]

4 Feb 2021

PONE-D-20-28926R1 

Not by the light of the moon: investigating circadian rhythms and environmental predictors of calling in Bornean great argus 

Dear Dr. Clink:

I'm pleased to inform you that your manuscript has been deemed suitable for publication in PLOS ONE. Congratulations! Your manuscript is now with our production department. 

Kind regards, 

on behalf of

Dr. Dennis M. Higgs 

Academic Editor

PLOS ONE